# Impact of mobile health applications on self-management in patients with type 2 diabetes mellitus: protocol of a systematic review

Benard Ayaka Bene,[1,2] Siobhan O'Connor,[3] Nikolaos Mastellos,[1] Azeem Majeed,[1] Kayode Philip Fadahunsi,[1,4] John O'Donoghue[5]

[1]Department of Primary Care & Public Health, School of Public Health, Imperial College London, London, UK
[2]Department of Public Health, Federal Ministry of Health, Abuja, Nigeria
[3]School of Health in Social Science, University of Edinburgh, Edinburgh, UK
[4]Department of Hospital Services, Federal Ministry of Health, Abuja, Nigeria
[5]Malawi eHealth Research Centre, University College Cork, Cork, Ireland

**Correspondence to**
Dr Benard Ayaka Bene;
b.bene15@imperial.ac.uk

## ABSTRACT

**Introduction** The emergence of mobile health (mHealth) solutions, particularly mHealth applications (apps), has shown promise in self-management of chronic diseases including type 2 diabetes mellitus (T2DM). While majority of the previous systematic reviews have focused on the effectiveness of mHealth apps in improving treatment outcomes in patients with T2DM, there is a need to also understand how mHealth apps influence self-management of T2DM. This is crucial to ensure improvement in the design and use of mHealth apps for T2DM. This protocol describes how a systematic review will be conducted to determine in which way(s) mHealth apps might impact on self-management of T2DM.

**Methods** The following electronic databases will be searched from inception to April 2019: PubMed, MEDLINE, EMBASE, Global Health, PsycINFO, CINAHL, The Cochrane Central Register of Controlled Trials, Scopus, Web of Science, ProQuest Dissertations & Theses Global, Health Management Information Consortium database, Google Scholar and ClinicalTrials.gov. The Cochrane risk of bias tool will be used to assess methodological quality. The primary outcome measures to be assessed will be 'change in blood glucose'. The secondary outcomes measures will be 'changes in cardiovascular risk markers' (including blood pressure, body mass index and blood lipids), and self-management practices. Others will include: health-related quality of life, economic data, social support, harms (eg, death or complications leading to hospital admissions or emergency unit attendances), death from any cause, anxiety or depression and adverse events (eg, hypoglycaemic episodes).

**Ethics and dissemination** This study will not involve the collection of primary data and will not require ethical approval. The review will be published in a peer-reviewed journal and a one-page summary of the findings will be shared with relevant organisations. Presentation of findings will be made at appropriate conferences.

**Trial registration number** CRD42017071106.

## Strengths and limitations of this study

► This study will extend its focus beyond assessing effectiveness in improving treatment outcomes to understanding how mobile health (mHealth) applications (apps) might influence self-management of type 2 diabetes mellitus (T2DM).

► The methodological quality of all included trials in this study will be thoroughly assessed in order to ascertain the validity of their findings.

► Robust subgroup analyses will provide an understanding of how certain factors or patient characteristics (such as ethnicity and presence of comorbidities) might affect self-management of T2DM when using mHealth apps.

► A wide range of databases will be searched to ensure that potentially relevant studies are not missed.

► Since only studies published in English language will be considered for this review, this might introduce some bias. However, we are aware that studies with significant findings are likely to be published in English language so as to increase their chances of being cited by others.

## INTRODUCTION

Diabetes is a long-term condition and a leading cause of morbidity and mortality world-wide.[1] The past three decades have seen the most dramatic increase in the number of adults living with diabetes by almost a four-fold, from 108 million in 1980 to 422 million in 2014.[2] Type 2 diabetes mellitus (T2DM), the most common type of diabetes in adults, accounts for over 90% of all diabetes cases.[1 3] When T2DM is poorly managed, it can easily result in systemic complications such as coronary heart disease, stroke, kidney failure, retinopathy and foot ulcers.[4] These complications can further progress to severe disabilities. For example, diabetic foot ulcers can lead to non-traumatic limb amputation and diabetic retinopathy can result in blindness.[4] Complications and disabilities resulting from poorly managed T2DM often cause increased socio-economic burden with associated reduced quality of life and reduced life expectancy.[5 6] A landmark study estimated the cost of type 2

diabetes mellitus in the UK in 2010/2011 at £8.8 billion in direct costs and £13 billion in indirect costs.[7] The severity of the burden of T2DM has further heightened the need to improve its treatment and management.

The treatment of T2DM primarily aims to control blood glucose thereby preventing or reducing associated complications and disabilities.[6] Over the years, there has been a growing body of evidence to support the role that self-management plays in the treatment of T2DM.[8–12] Self-management is a term used to describe patient's own responsibilities (including practices and skills) employed in maintaining good health.[13 14] The documented practices and skills which form critical components of the management of T2DM are mainly healthy eating, physical activity, blood sugar monitoring, medication adherence, good problem-solving skills, healthy coping skills and risk-reduction behaviours.[10 11 13 14]

Mobile health (mHealth) solutions, which include mobile applications (apps), have been rapidly gaining popularity in the management of chronic diseases and have further created opportunities and potential to enhance the ability of T2DM patients for self-management.[15–17] A mobile app is a software application designed to run on smartphones, tablet computers or similar mobile devices.[18] When mobile apps are used for health purposes, they are often referred to as mHealth apps. They have the ability to facilitate one or more aspects of self-management by capturing user's health data and providing tailored information, instructions, graphic displays, guidance and reminders to users.[18–20] In addition, mHealth apps are designed with aesthetical features to appeal to users and can provide a portable platform for remote monitoring of patient's data as well as links to their healthcare providers and social networks.[18–21] More specifically, the definition of mHealth app for self-management of T2DM in the context of this study is adapted from Pal *et al*[19] as any mobile application which utilises input from a patient by means of communication or processing technology to provide tailored responses that facilitate one or more aspect of self-management of T2DM (healthy eating, physical activity, blood sugar monitoring, medication adherence, good problem-solving skills, healthy coping skills and risk-reduction behaviours).[19]

Although mHealth apps seem promising for influencing self-management of T2DM,[22] concerns have been raised about their quality and safety following evaluation studies which showed that some of these apps are either poorly designed, do not function as intended or do not adhere to evidence-based guidelines.[20 21 23 24] While previous systematic reviews showed modest benefits of mHealth apps in self-management of T2DM, they focused on assessing effectiveness in improving treatment outcomes rather than understanding how these mHealth apps most effectively influence self-management of T2DM.[16 19 25–28] The use of mHealth apps, especially in the context of self-management, is a complex intervention (influenced by several interacting components including healthy eating, physical activity, blood

sugar monitoring, medication adherence, good problem-solving skills, healthy coping skills and risk-reduction behaviours).[29] Therefore, extending the focus beyond assessing effectiveness to understanding how (including when and where) mHealth apps influence self-management of T2DM is extremely important. This will provide evidence and direction for better design, implementation and ultimately, the optimum use of mHealth apps for self-management of T2DM.

In this article, we present a protocol which describes how a systematic review will be conducted to determine in what way(s) mHealth apps might impact on self-management of T2DM and thus provide an additional perspective on how (including when and where) mHealth apps may influence self-management of T2DM. The protocol is presented in accordance with the guideline of the Preferred Reporting Items for Systematic Review and Meta-Analysis Protocol (PRISMA-P).[30] A completed PRISMA-P checklist is provided as online supplementary file 1.

## AIM AND RESEARCH QUESTION
The aim is to determine how mHealth apps might impact on self-management of T2DM. The review will attempt to answer a crucial research question, which to the best of our knowledge has not been fully answered by previous systematic reviews; that is, how does the use of mHealth apps impact on self-management of T2DM in patients compared with other interventions?

## METHODS
### Study design
A team comprising of experts from the relevant disciplines (diabetes management, information and communication technologies and systematic review methodology) will design, conduct and report the systematic review. The formation of the review question and search strategy was guided by the Participants, Intervention, Comparison, Outcomes framework.[31 32] The process of the systematic review will follow the methods described in the Cochrane Handbook for Systematic Reviews of Interventions.[33] The reporting of the review will be guided by the Preferred Reporting Items for Systematic Review and Meta-Analysis (PRISMA) statement.[34]

### Study registration
This systematic review is registered with PROSPERO ( www.crd.york.ac.uk/PROSPERO). Registration number: CRD42017071106.

### Criteria for considering studies for this review
#### Type of studies
Only randomised controlled trials (RCTs) will be included in this review with no restriction in the duration of follow-up. The Consolidated Standards of Reporting Trials checklist will be used to judge the reliability or relevance of the findings of RCTs that will be included

in this review.[35] The risk of bias will be assessed using the Cochrane Collaboration's tool.[33]

## Types of participants

Patients diagnosed with T2DM will be considered for this review. Studies that included both type 1 and type 2 diabetes patients will also be considered; however, only data on patients with type 2 diabetes will be extracted. Studies targeted at only patients with type 1 diabetes will not be considered. There will be no age restriction, but participants will be categorised by age group: ≤39 years, 40 to 65 years and >65 years. Older patients are likely to have more diabetes comorbid conditions (such as raised blood pressure) than younger patients,[6] while younger patients are likely to be more digitally literate and thus more inclined to use mHealth.[36]

## Diagnostic criteria for T2DM

T2DM is characterised by hyperglycaemia resulting from progressive insulin resistance and deficiency.[37] For consistency, the current WHO/International Diabetes Federation diagnostic criteria for diabetes will be maintained; that is, fasting blood glucose ≥7.0 mmol/L (126 mg/dL) or 2 hour blood glucose ≥11.1 mmol/L (200 mg/dL).[38] Where glycated haemoglobin (HbA1c) is used as a diagnostic criterion, the WHO recommended value of ≥6.5% will be used.[39] Where diagnostic criteria are not stated, authors will be contacted.

## Types of intervention

Only studies on self-management of T2DM that utilised mHealth apps alone, mHealth app along with usual care or mHealth apps along with a range of other technologies such as wearable devices (for example, pedometer) or mHealth apps in conjunction with other mHealth solutions such as texting or messaging will be included in this review. Studies that used mHealth solutions (such as emailing and texting) exclusively for communication between patients and health professionals or social networks; or targeted exclusively at health professionals will not be considered for this review as they provide limited functionality for self-management.

## Types of comparison/control

Comparisons will be made against any type of control. This may include, but not limited to, standard or usual care, dummy apps or control apps, face-to-face self-management education, use of paper educational materials, other mHealth solutions (for example, messaging or texting), computer-based and/or web-based self-management interventions.[40]

## Types of outcome measures

The outcome measures of this review will be reported as primary and secondary outcomes based on reported outcomes of included studies.

The primary outcomes will be 'change in blood glucose' often reported as glycated haemoglobin (HbA1c). HbA1c is the gold standard for assessing glycaemic control in diabetic patients and each measurement represents average blood glucose over the previous 2 to 3 months. HbA1c measurement does not require any special preparation such as fasting and it can be done at any time of the day.[38] If fasting blood glucose is reported rather than HbA1c in some included studies, it will then be considered as the primary outcome measure; however, it will be converted to an estimated HbA1c value.

The secondary outcomes will include 'changes in cardiovascular risk markers (blood pressure, body mass index, low-density lipoprotein cholesterol, high-density lipoprotein cholesterol and triglyceride), patient's knowledge on T2DM and self-management and adherence to self-management practices. Others will include: health-related quality of life, economic data (such as cost-effectiveness), social support, harms (such as death or complications leading to hospital admissions or emergency unit attendances), death from any cause, anxiety or depression and adverse events (for example, hypoglycaemic episodes).[40]

## Timing of outcome measurement

Where possible, the impact of the intervention at different timings will be measured. The timing will be grouped into three categories of follow-up as follows: short-term, medium-term and long-term. Short-term follow-up will be defined as that measured within $3^{3}$ months of the intervention period in order to determine the immediate changes resulting from the intervention. Medium-term follow-up will be defined as that measured between $3^{3}$ and $6^{6}$ months of the intervention period to determine if the changes continue. Long-term follow-up will be defined as $6^{6}$ months and over after the intervention to determine whether there are changes over time.[40] For the overall meta-analysis, the longest follow-up data available will be used.

## Search strategy for the identification of studies

Using the key terms (type 2 diabetes mellitus, self-management, mobile health, mHealth and mobile application), a comprehensive search strategy will be designed by two reviewers (BAB and SOC) with the assistance of a librarian and in consultation with other research team members. The search strategy will be used to search for all eligible studies including articles, dissertations, theses, conference proceedings and grey literature (including committee reports and government reports). Online trial registers for ongoing and recently completed studies will also be searched. While no restriction will be placed on dates, only studies reported in English language will be considered.

The following electronic databases will be searched from their inception to April 2019:

PubMed, MEDLINE, EMBASE, Global Health, PsycINFO, CINAHL, The Cochrane Central Register of Controlled Trials, Scopus, Web of Science, ProQuest Dissertations & Theses Global, Health Management Information Consortium database, Google Scholar and ClinicalTrials.gov.

Additional studies will be identified by searching the reference lists of included studies as well as reference list of relevant systematic reviews and meta-analyses.

A re-run of the entire searches will be done just before the final analyses and any additional studies found will be included.

A sample search strategy for MEDLINE is provided in online supplementary file 2.

## Selection of studies

All identified articles will be imported into Mendeley reference management software, and duplicates will be removed. The articles will then be imported into Covidence (a web-based tool to support the reviewers to manage the data). Two reviewers working independently will screen each article for possible inclusion in the review. The screening will be done in two stages (title and abstract, and full text) based on predefined eligibility criteria as highlighted in table 1. To ensure consistency in the screening process, the two reviewers (BAB and SOC) will pilot the entire process on 10 studies as guided by the Cochrane Collaboration Study Selection and Data Extraction form.[33] A consensus will be reached after discussing and refining the process. The reasons for excluding any study will be published with the main study. Any disagreement will be resolved by discussion and where there is an unresolved disagreement, a third party (JOD) will be invited to resolve the issue which will be justified in a steering group meeting. The entire selection processes will be described using the Preferred Reporting Items for Systematic Reviews and Meta-Analyses (PRISMA) flow diagram.[34] The PRISMA checklist will be completed and attached as an additional file.

## Data extraction and management

Two reviewers (BAB and SOC) working independently will extract the characteristics of selected studies using standard data extraction templates as guided by the Cochrane Collaboration Study Selection and Data Extraction form.[33] Any disagreement will be resolved by discussion. Where there are inconsistencies or unresolved disagreements, a third party (JOD) will be invited to resolve the issue which will be justified in a steering group meeting. To ensure consistency in the extraction process, it will be initially piloted on at least 10[10] per cent of the articles and a consensus reached after discussing and refining the process. Any missing information that is relevant to this review will be sought from the original authors of the article by email.

The following characteristics will be included if reported in individual studies[41]:

► Publication details: authors, year and country of study.
► Methods: study design, baseline measure, time points (when data were collected: at baseline and endpoint) and study setting (location, year and environment).
► Participant characteristics: number of participants, mean age or age range, gender ratio, ethnicity, socioeconomic group, educational status, duration of

T2DM and participant inclusion criteria and exclusion criteria.

► Intervention: description of the content and functions design of the mHealth apps used, the aspects of self-management, number of participants allocated to the intervention group, other technologies or interventions used and duration.
► Control/comparison(s) group: description of the comparison(s) and number of participants allocated to the control group.

| Table 1 | Predefined criteria for inclusion in the systematic review | |
|---|---|---|
| **Acronym** | **Term** | **Description** |
| P | Population | Patients with T2DM as defined by WHO & IDF diagnostic criteria.[38][39] |
| I | Intervention | Studies on self-management of T2DM that utilised mHealth apps alone, mHealth apps along with usual care or along with a range of other technologies such as a wearable device (eg, pedometer) or mHealth apps in conjunction with other mHealth solutions such as texting/messaging. |
| C | Comparison | The control groups be used for comparison. These may include standard or usual care, dummy apps or control apps, face-to-face self-management education, use of paper educational materials, other mHealth solutions (for example, messaging or texting), computer-based and/or web-based self-management interventions. |
| O | Outcomes | Primary outcomes will be change in blood glucose (HbA1c). The secondary outcomes will include changes in cardiovascular risk markers (BP, BMI, LDL-C, HDL-C and TG), patient's knowledge on T2DM and self-management, and adherence to self-management practices. Others will include health-related quality of life, economic data (such as cost-effectiveness), social support, harms (such as death or complications leading to hospital admissions or emergency unit attendances), death from any cause, anxiety or depression and adverse events (eg, hypoglycaemic episodes). |
| S | Study type | Randomised controlled trials. |
| T | Timing of outcome measure | There will be no restriction to the timing of outcome measures, however, the timing will be grouped into three categories: short-term (≤3 months of the intervention period), medium-term (3 to 6 months of the intervention period and long-term (≥6 months after the intervention). |

apps, applications; BP, blood pressure; BMI, body mass index; HbA1c, glycated haemoglobin; HDL-C, high-density lipoprotein cholesterol; IDF, International Diabetes Federation; LDL-C, low-density lipoprotein cholesterol; mHealth, mobile health; TG, triglyceride; T2DM, type 2 diabetes mellitus.

- ► Outcomes: description of primary, secondary and other outcomes, list of measurement tools and devices, unit of measurement for outcomes and intervention effects on the outcomes (effect size, 95% CI, standard mean deviation).
- ► Additional information: any information that may express conflict of interest or bias will be noted.

### Assessment of risk of bias in included studies

Each study will be assessed independently by two reviewers (BAB and NM). Any disagreements will be resolved by discussion, or if required, a third party (JOD).

The following bias criteria will be used to assess the risk of bias as recommended in the Cochrane Handbook for Systematic Reviews of Interventions[33]:

- ► Random sequence generation (selection bias).
- ► Allocation concealment (selection bias).
- ► Blinding (performance bias and detection bias), separated for blinding of participants and personnel and blinding of outcome assessment.
- ► Incomplete outcome data (attrition bias).
- ► Selective reporting (reporting bias).
- ► Other bias.

The risk of bias criteria for RCTs will be judged as 'low risk', 'high risk' or 'unclear risk' and the use of individual bias items as described in the Cochrane Handbook for Systematic Reviews of Interventions.[33] A 'risk of bias graph' figure and 'risk of bias summary' figure will be attached. The impact of individual bias domains on study results at endpoint and study levels will be assessed.

## DATA SYNTHESIS

Both qualitative and quantitative analyses are planned for this review.

### Qualitative synthesis

For the qualitative analysis of this review, a narrative synthesis approach will be adopted based on the Guidance on the Conduct of Narrative Synthesis in Systematic Reviews.[42] Popay *et al* defined narrative synthesis as 'an approach to the systematic review and synthesis of findings from multiple studies that relies primarily on the use of words and text to summarise and explain the findings of the synthesis'.[42]

Narrative synthesis approach is adopted for this review so as to develop a preliminary synthesis, explore relationships within and between studies and assess the robustness of the synthesis.[42] In preliminary synthesis, the results of included studies are laid out in a systematic manner to give an overview of the relationships among them allowing for comparison of direction and size of effects, which will be further explored in the next step. The next step involves examining the relationships within and between studies categorising and explaining factors responsible for the differences in direction and effects as well as the interplay of factors that may influence effectiveness and successful implementation. Finally, the entire process of

narrative synthesis allows for the methodological quality of included studies to be scrutinised thereby increasing the robustness of the review.

### Quantitative synthesis

Statistical analyses will be performed based on recommendation in the Cochrane Handbook for Systematic Reviews of Interventions.[43] Summaries of intervention effects for each study will be calculated using risk ratios (for dichotomous outcomes) or standardised mean differences (for continuous outcomes). For meta-analysis, it is anticipated that there will be limited scope for the use of fixed-effect model because of the possibility of a range of different outcome measures and also, the effect sizes are not likely to be identical across studies.[44] For instance, the magnitude of the impact of mHealth apps alone or along with other technologies (such as wearable devices) or in conjunction with other interventions on self-management might vary. Therefore, random-effects model will be used as the weights assigned under random effects are more balanced.[44]

### Measures of treatment effect
#### Dichotomous data

The effect size for dichotomous data will be expressed as risk ratios and 95% CI. The risk difference will be calculated as well as the number needed to treat for an additional beneficial outcome or the number needed to treat for an additional harmful outcome, when possible.

#### Continuous data

For continuous outcomes, weighted mean differences and 95% CI will be calculated. If results for some continuous outcomes are found on different scales and cannot be converted to a standard scale standardised mean differences will be used.

#### Time-to-event data

The results will be expressed as HR with corresponding 95% CI.

#### Unit of analysis issues

The review will take into account the level at which randomisation occurred, such as cross-over trials, cluster-randomised trials and multiple observations for the same outcome.

### Dealing with missing data

Relevant missing data will be obtained from original authors if feasible and an evaluation of important numerical data such as numbers of screened articles, randomised patients, intention-to-treat, as-treated and per-protocol population will be done. Attrition rates, for example dropouts, losses to follow-up and withdrawals will be investigated and issues of missing data and imputation methods (for example, last observation carried forward) will be critically appraised.

## Assessment of heterogeneity

In the event of substantial clinical or methodological or statistical heterogeneity, report of study results will not be presented as pooled effect estimates. Heterogeneity will be identified by visual inspection of the forest plots and by using a standard $X^2$ test with a significance level of $\alpha=0.1$, in view of the low power of this test. Specifically, heterogeneity will be examined by employing the $I^2$ statistic which quantifies inconsistency across studies to assess the impact of heterogeneity on the meta-analysis,[45 46] where an $I^2$ statistic of 75% and more indicates a considerable level of inconsistency.[43] When heterogeneity is found, an attempt will be made to determine potential reasons for it by examining individual study and subgroup characteristics. This will be reported as qualitative analysis using narrative synthesis.

## Assessment of reporting biases

To assess small study bias, funnel plots will be used if more than 10 studies are included for a given outcome.

## Subgroup analyses

Subgroup analyses will be performed for the purpose of assessing whether or not there exist any differences in the primary outcome influenced by certain factors or patient characteristics; however, there are scepticisms about the credibility of subgroup effects.[47–49] Therefore, we will ensure that subgroup analyses are conducted majorly if the primary outcome of any included trial shows statistically significant differences between intervention groups. Where a trial reports differences in treatment outcome between intervention groups but fails to demonstrate any statistical significance, subgroup analyses will only be carried out to generate hypotheses.[49] Thus, the following subgroup analyses are planned:

► Ethnicity/country of origin: An American study compared Hispanics with non-Hispanic Whites, who participated equally in a diabetes education class, and found that Hispanics were less likely to check their blood glucose daily or examine their feet for any abnormality. They were, however, more likely to take oral hypoglycaemic agents than non-Hispanic White.[50] Another study showed that Chinese Americans were more engaged than African Americans in improving most self-management behaviours.[51] We will perform a subgroup analysis to see the effect of ethnicity on self-management of T2DM when using mHealth apps.

► Comorbidities: A study found that diabetes patients who had higher number of comorbidities placed lower priority on their disease and hence scored low in their self-management ability.[52] This is likely to affect blood glucose control. Our study will attempt to find out if this hypothesis holds true for self-management of T2DM when using mHealth apps.

► Behaviour change model used: Technology-based interventions for diabetes have the potential to improve self-management; however, it has been argued that in order to achieve the desired patient benefit or treatment outcome, their design must be guided by behaviour change or self-care theories.[53] We will carry out a subgroup analysis to find out if this hypothesis also holds for mHealth apps for self-management of T2DM.

## Sensitivity analyses

Sensitivity analyses will be performed in order to explore the influence of the following factors on effect size:

► Restricting the analysis to published studies (RCTs).
► Restricting the analysis taking account risk of bias, as specified above.
► Restricting the analysis to long (≥12 months) or studies with relatively larger sample sizes to establish how much they dominate the results.
► Restricting the analysis to studies using the following filters: diagnostic criteria, source of funding (industry vs other) and country.

The robustness of the results will be tested by repeating the analysis using different measures of effect size (relative risk, odd ratio etc) and different statistical models (fixed-effect model and random-effects model).

## PATIENT AND PUBLIC INVOLVEMENT

Although patients and the public were not directly involved in the design of this study, the development of the research question was primarily informed by patients' interests in the research outcomes.

## ETHICS AND DISSEMINATION

This study does not involve collection of primary data from patients, hence it will not require ethical approval.

A manuscript will be submitted to a peer-reviewed journal for publication. Likewise, a summary of the findings will be shared with relevant and responsible organisations. In addition, important findings will be summarised and presented at national and international conferences such as the Diabetes UK Annual Scientific Meeting, and Society for Academic Primary Care National Meeting.

## DISCUSSION

The use of mHealth apps for self-management is a complex intervention because of the several interacting components involved (including healthy eating, physical activity, blood sugar monitoring, medication adherence, good problem-solving skills, healthy coping skills and risk-reduction behaviours). Hence, improving the design and use of mHealth apps for self-management of T2DM will require an understanding of how mHealth apps are likely to be most effective in influencing self-management of T2DM. The majority of previous studies primarily assessed the effectiveness of mHealth apps in improving health outcomes,[16 25 26 28] but this study will extend its focus to understanding how (including when and where) mHealth apps might influence self-management of T2DM. We will perform subgroup

analyses to assess any differences in the primary treatment outcome based on certain factors or patient characteristics such as ethnicity and the presence or absence of comorbidities. However, where a trial report suggests differences in treatment outcome between intervention groups but fails to demonstrate any statistical significance, subgroup analyses will only be carried out to generate hypotheses.

To our knowledge, this is the first published protocol that describes how a systematic review will be conducted to evaluate the impact mHealth apps might have on self-management of T2DM. In addition, this review will ensure robust assessment of methodological quality of included trials in order to ascertain the validity of their findings and to ensure that the risks of bias were minimised.[33 54]

In most of the previous systematic reviews, limited databases were searched. For instance, Cui et al[26] and Liang et al[16] searched three databases while Frazetta et al[28] searched two databases.[16 26 28] In this review however, a wide range of databases will be searched to ensure that potentially relevant studies are not missed. Although only studies published in English language will be considered for this review, we are cognisant of the fact that studies with significant findings are likely to be published in English language so as to increase their chances of being cited by others.[55]

Finally, it is expected that the evidence which will be generated from this study will add a new perspective that will be useful in informing improvement and/or optimisation of design and use of mHealth apps for self-management of T2DM; thus, potentially improving health outcomes in patients with T2DM.

**Acknowledgements** We sincerely appreciate the assistance of Rebecca Jones, the Library Manager and Liaison Librarian at the Charing Cross Library, Imperial College London, with developing the search strategy for this review.

**Contributors** BAB and JOD conceived the study. JOD, NM, SOC, AM and KPF contributed to the study design and methodology. SOC and KPF specifically contributed to the keywords and search strategy. BAB drafted the manuscript and all the research team members contributed significantly to it. AM is the clinical lead while JOD acts as guarantor for the study. The final manuscript was read and approved by all the authors.

**Funding** This article presents independent research in part funded by the National Institute for Health Research (NIHR) under the Collaborations for Leadership in Applied Health Research and Care (CLAHRC) programme for North West London. The views expressed in this publication are those of the author(s) and not necessarily those of the NHS, the NIHR or the Department of Health and Social Care.

**Competing interests** None declared.

**Patient consent for publication** Not required.

**Provenance and peer review** Not commissioned; externally peer reviewed.

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
