## [Reviewer comments · BMJ Open]

ARTICLE DETAILS

TITLE (PROVISIONAL)	The Impact of Mobile Health Applications on Self-Management in Patients with Type 2 Diabetes Mellitus: Protocol of a Systematic Review
AUTHORS	Bene, Benard; O'Connor, Siobhan; Mastellos, Nikolaos; Majeed, Azeem; Fadahunsi, Kayode; O'Donoghue, John

VERSION 1 - REVIEW

REVIEWER	Sheyu Li West China Hospital, Sichuan University
REVIEW RETURNED	22-Aug-2018

GENERAL COMMENTS	Bene and colleagues presented a protocol for systematic review and meta-analysis of mobile health application on the self-management in adults with type 2 diabetes. The topic is important. The protocol of the systematic review has been comprehensively described. But as many systematic reviews of this topic have been published, the effectiveness of the mobile applications in patients with diabetes is almost confirmed. The authors need to further highlight what will be added after the current systematic review added. And what will be the main difference between this systematic review and the previous one? I have some concerns before its publication: 1. The manuscript could be shortened and more clear.2. If the authors think the previous systematic reviews paid little attention to the quality assessment of the included trials, they would better provide some evidence. Meanwhile, a comprehensive reviewing of the previous studies should be conducted.3. When describing the inclusion criteria for the study type, the authors could simply state that they include both RCTs and observational studies, and exclude the non-randomized trials. Meanwhile, non-randomized studies could be an overlapping concept with observational studies. The authors may wish to use the term of non-randomized trials. The observational studies included cross-sectional studies, case-control studies, cohort studies, single-arm case series and so on. I assume that the authors aimed to include the cohort studies only. Is that right? If the authors planned to exclude observational studies with the follow-up duration of less than 12 months, should they exclude the trials with shorter follow-up duration?
--

	4. The criteria of participants were confused. It seemed to be criteria for an original study rather than a systematic review. What if a study included patients older than 14 years old or they adopt both type 1 and type 2 diabetes at the same time? Actually, many trials included the mixed population. Additional bias can be introduced if all these trials are excluded. 5. Searching for a number of literature databases seems to be a strength stated by the authors in the introduction section. However, are the authors considering to search some database of trial registration website or congress abstract? BTW, the searching strategy could be moved to the supplementary data. 6. We use PICOST rather than PICO currently when S stands for study type and T stands for time, which is follow-up duration. 7. It is not reasonable to pool trials and observational studies in a single meta-analysis. The authors need to state it clearly. 8. There are too many subgroup analyses to introduce excessive type I error, and most of them seem to be inapplicable for a systematic review. Please restrict the number of subgroup analyses due to the expected number of included studies. And make sure all the planned subgroup analyses could be done after you include the studies. 9. Please forget the first sensitivity analysis because RCTs should not be pooled together with observational studies. 10. For the second sensitivity analysis, the authors may wish to use the quality-effect model to pool the results. 11. Please clearly define what 'very long' and 'very large' mean in the third sensitivity analysis. 12. An expected result of the systematic review and its potential use in the clinical practice could be briefly discussed before the conclusion section. 13. The expected strength and limitation of the study and the difference between the current systematic review and previous ones could be added in the Discussion section.
--	---

REVIEWER	Rosie Dobson University of Auckland, New Zealand
REVIEW RETURNED	11-Oct-2018

GENERAL COMMENTS	The authors should be congratulated on an excellent protocol paper for a systematic review which will be a good contribution to the research literature. This protocol describes a systematic review and meta-analysis of mHealth apps for supporting self-management in people with type 2 diabetes. This is an important area, as the field of mHealth continues to advance and apps become more widely available understanding the evidence for these is important for guiding decisions to recommend or use them in practice. Specific comments:  - The key words should be updated to include more specific terms related to the study i.e. mHealth, apps - The introduction could be strengthened with more evidence for the potential of mHealth and specifically apps in diabetes self-management. Currently there is just one sentence "Mobile health (mHealth) solutions, which include mobile applications (apps), have been rapidly gaining popularity in the management of chronic diseases and have further created opportunities and potentials for T2DM patients to gain knowledge and skills for self-management
---

(15–17).” Which I do not believe adequately introduces the role of apps in self-management of diabetes. It would be good to cover the types of apps and functionalities that they provide, for example blood glucose monitoring, education, insulin calculations. How do the authors define what is a self-management app for diabetes? You could argue that any apps that target a behaviour of diabetes self-management are (i.e. the fitbit app for exercise) are but do they need to be diabetes specific (which the fitbit one is not)? I also think it would be beneficial to comment on the fact there has been a lot of criticism in the literature about the quality of health apps including those in diabetes i.e. Huckvale 2015 (this provides further justification for the review).

- It would be good to comment on why the authors have limited the review to adults. In the context of rising rates of T2D in young people globally and that younger people may be more receptive to mHealth (as the authors comment on) it would make sense to not limit by age. Also please note that in some countries in healthcare an adult is defined as 16 years and older and I would hope that the authors would not exclude studies of adults using the definition or >16 years.

- Types of participants:

o The authors need to specify that they will include studies with both T2DM and type 1 as this is not clear.

o Please see previous comment in regards to age

- Types of intervention:

o The first part of this section may be better placed in the introduction to the paper as per my second point above.

o The authors may need to specify a definition of what constitutes a self-management app. They state that they will not include apps for communication with healthcare professionals which is still part of self-management of diabetes so therefore what about an app that just provides a log BG values sent directly via Bluetooth from a BG monitor, or what about calorie counting apps that are just reference tools, or what about apps for tracking appointments with care team? Apps can vary considerably - basic education, behaviour tracking, reference sources, behaviour change intervention, insulin calculators, or data repositories. Are there any limitations of these or requirements to meet the criteria of a self-management app? Do they need to provide a component of self-management education or be related to a specific self-management behaviour?

- Types of comparison/control:

o Will you include control apps? I.e. apps that provide limited or dummy intervention which are common in RCTs of app interventions.

- Types of outcome measurement:

o Should the primary outcome be “changes in blood glucose control” not “changes in blood glucose”

- Timing of outcome measurement:

o The authors have stated that HbA1c will be a primary outcome. HbA1c is an average measure of glycaemic control typically over 2-3 months. But the authors have said they will define ‘short-term follow up’ as less than one month. This does not make sense in relation to the primary outcome measure. It would make more sense to me that short term was less than 3 months if you are using glycaemic control as a primary outcome.

- Search strategy for the identification of studies:

o Do you mean “key terms” not key concepts”?

o Is there any reason you have excluded Google Scholar and Web of Science from the list of databases? The authors state that a

	limitation of previous reviews is the limited databases searched so it is important that authors ensure that have overcome this by including all relevant databases. Considering the types of journals that mHealth studies are typically published in, many of which are not indexed in places like Scopus, google scholar may be an important source of studies for this review. - Table 2:  o Intervention: What about mHealth + usual care? o Comparison: see previous comment above regarding whether you will include control apps - Quantitative synthesis: How do the authors plan to deal with the 2 different measurements of the primary outcome? HbA1c and FBG? How will you compare studies that use the different outcome measures?
--	--

VERSION 1 – AUTHOR RESPONSE

Reviewer 1 Comment	Author Response	Page No
The authors need to further highlight what will be added after the current systematic review. And what will be the main difference between this systematic review and the previous one?	Thank you for the comment. The main differences between the current systematic review and previous ones are now highlighted in the introduction section and further summarised in the discussion section:  • Previous studies focused on assessing effectiveness; however, this study will extend its focus to understanding how (including when and where) mHealth apps might most effectively influence self-management of T2DM. • Unlike most previous systematic reviews, the methodological quality of all included trials in this study will be thoroughly assessed in order to ascertain the validity of their findings. • A robust subgroup analysis will provide an understanding of the influence of various factors including demographics (such as gender, age, ethnicity and social status) on mHealth app interventions for self-management of T2DM. • While limited databases were searched in previous systematic reviews, this review will ensure search of a wider range of 	3-5, 13- 14

	databases so as not be miss potentially relevant studies. The strengths of the review are more clearly outlined in the 'Strengths and Limitations' section as well as in the 'Discussion' section.	
The manuscript could be shortened and more clear.	Unnecessary texts removed and sentences reconstructed to shorten and make manuscript more concise.	All
If the authors think the previous systematic reviews paid little attention to the quality assessment of the included trials, they would better provide some evidence. Meanwhile, a comprehensive reviewing of the previous studies should be conducted.	The section has been modified as thus: "Previous systematic reviews paid too little attention to the assessment of methodological quality of included trials. For instance, Cui et al 2016 assessed the quality of included studies using the Cochrane Collaboration's tool, but limited detail was reported; while Liang et al 2011 and Frazetta et al 2012 did not report any information on methodological quality assessment of included trials (1–3). This review will ensure robust assessment of methodological quality of included trials in order to ascertain the validity of their findings and to ensure that the risks of bias are minimised (4,5)". The Consolidated Standards of Reporting Trials (CONSORT) statement will be used to judge the reliability or relevance of the findings of all included randomised controlled trials (RCT) (6). The risk of bias will be assessed using the Cochrane Collaboration's tool (5)	13 – 14
When describing the inclusion criteria for the study type, the authors could simply state that they include both RCTs and observational studies, and exclude the non-randomized trials. Meanwhile, non-randomized studies could be an overlapping concept with observational studies. The authors may wish to use the term of non-randomized trials. The observational studies included cross-sectional studies, case-control studies, cohort studies, single-arm case series and so on. I assume that the authors aimed to include the cohort studies only. Is that right? If the authors planned to exclude observational studies with the follow-up duration of less than 12 months, should they exclude the trials with shorter follow-up duration?	After due consideration, we decided to restrict study type to RCTs only, but with no restriction in the duration of follow-up	6, 9
The criteria of participants were confused. It seemed to be criteria for an original study rather	We acknowledge that more younger persons are now diagnosed with T2DM.	6, 8

than a systematic review. What if a study included patients older than 14 years old or they adopt both type 1 and type 2 diabetes at the same time? Actually, many trials included the mixed population. Additional bias can be introduced if all these trials are excluded.	Therefore, age restriction has been removed. Studies that included both type 1 and type 2 diabetes will be considered for the review. However, only data from type 2 diabetes patients will be extracted.	
Searching for a number of literature databases seems to be a strength stated by the authors in the introduction section. However, are the authors considering to search some database of trial registration website or congress abstract? BTW, the searching strategy could be moved to the supplementary data.	Yes, ClinicalTrials.gov and Cochrane Central Register of Controlled Trials [CENTRAL] are included to the databases to be searched. The search strategy has now been moved to supplementary data.	7
We use PICOST rather than PICO currently when S stands for study type and T stands for time, which is follow-up duration.	Study type (S) and time (T) are included in the predefined criteria for inclusion of studies.	8 – 9
It is not reasonable to pool trials and observational studies in a single meta-analysis. The authors need to state it clearly.	Only RCTs are now being considered for this review.	6, 9
There are too many subgroup analyses to introduce excessive type I error, and most of them seem to be inapplicable for a systematic review. Please restrict the number of subgroup analyses due to the expected number of included studies. And make sure all the planned subgroup analyses could be done after you include the studies.	The planned subgroup analysis has been reduced from 13 to 8.	12
Please forget the first sensitivity analysis because RCTs should not be pooled together with observational studies.	The review will now include only RCTs.	12
For the second sensitivity analysis, the authors may wish to use the quality-effect model to pool the results.	Yes, thank you.	12
Please clearly define what 'very long' and 'very large' mean in the third sensitivity analysis.	The statement is rephrased to read: "Restricting the analysis to long (≥ 12 months) or studies with relatively larger sample sizes to establish how much they dominate the results.	12
An expected result of the systematic review and its potential use in the clinical practice could be briefly discussed before the conclusion section.	Yes, this is now included in the discussion section.	13 – 14
The expected strength and limitation of the study and the difference between the current systematic review and previous ones could be added in the Discussion section.	Yes, the strength and limitation of this review as well as the differences between the current review and previous reviews have been added to the discussion section.	13 – 14
Reviewer 2 Comment	Author Response	Page No

The key words should be updated to include more specific terms related to the study i.e. mHealth, apps.	The key words are changed to: systematic review, mHealth, apps, self-management, type 2 diabetes mellitus.	2
The introduction could be strengthened with more evidence for the potential of mHealth and specifically apps in diabetes self-management. Currently there is just one sentence “Mobile health (mHealth) solutions, which include mobile applications (apps), have been rapidly gaining popularity in the management of chronic diseases and have further created opportunities and potentials for T2DM patients to gain knowledge and skills for self-management (15–17).” Which I do not believe adequately introduces the role of apps in self-management of diabetes. It would be good to cover the types of apps and functionalities that they provide, for example blood glucose monitoring, education, insulin calculations. How do the authors define what is a self-management app for diabetes? You could argue that any apps that target a behaviour of diabetes self-management are (i.e. the fitbit app for exercise) are but do they need to be diabetes specific (which the fitbit one is not)? I also think it would be beneficial to comment on the fact there has been a lot of criticism in the literature about the quality of health apps including those in diabetes i.e. Huckvale 2015 (this provides further justification for the review).	The introduction has been strengthened by including how mHealth apps are applicable for self-management; and definition and role of mHealth apps in self-management of T2DM. Statement on concerns about the safety of some mHealth apps as identified by other authors has also been included, which further justifies the need for this review.	4-5
It would be good to comment on why the authors have limited the review to adults. In the context of rising rates of T2D in young people globally and that younger people may be more receptive to mHealth (as the authors comment on) it would make sense to not limit by age. Also please note that in some countries in healthcare an adult is defined as 16 years and older and I would hope that the authors would not exclude studies of adults using the definition or >16 years.	The restriction on age has been removed. As you rightly pointed out, there is a rising rate of T2DM among young people globally.	6
Types of participants: o The authors need to specify that they will include studies with both T2DM and type 1 as this is not clear. o Please see previous comment in regards to age.	We have now specified that studies which included both type 1 and type 2 diabetes will be considered. However, only data from type 2 diabetes patients will be extracted.	

Types of intervention:  o The first part of this section may be better placed in the introduction to the paper as per my second point above. 	The first part of this section us moved to the introduction section.	4 – 5
 o The authors may need to specify a definition of what constitutes a self-management app. They state that they will not include apps for communication with healthcare professionals which is still part of self-management of diabetes so therefore what about an app that just provides a log BG values sent directly via Bluetooth from a BG monitor, or what about calorie counting apps that are just reference tools, or what about apps for tracking appointments with care team? Apps can vary considerably - basic education, behaviour tracking, reference sources, behaviour change intervention, insulin calculators, or data repositories. Are there any limitations of these or requirements to meet the criteria of a self-management app? Do they need to provide a component of self-management education or be related to a specific self-management behaviour? 	The definition of what constitutes a self-management app has been highlighted in the introductory section. Any mHealth solution that is used exclusively for communication purposes (such as emailing and texting) are not classified as an mHealth app for self-management. The definition of mHealth app for self-management of T2DM in the context of this study is adapted from Pal et al (2014) as any mobile application which utilises input from a patient by means of communication or processing technology to provide tailored responses that facilitate one or more aspect of self- management of T2DM (healthy eating, physical activity, blood sugar monitoring, medication adherence, good problem-solving skills, healthy coping skills and risk-reduction behaviours) (7).	4 – 5
Types of comparison/control:  o Will you include control apps? I.e. apps that provide limited or dummy intervention which are common in RCTs of app interventions. 	Yes, comparisons will include control apps or dummy apps.	8, 11
Types of outcome measurement:  o Should the primary outcome be “changes in blood glucose control” not “changes in blood glucose” 	The primary outcome will be “change in blood glucose”. This is the difference between the blood glucose at baseline and at end-point.	8, 12

Timing of outcome measurement:  o The authors have stated that HbA1c will be a primary outcome. HbA1c is an average measure of glycaemic control typically over 2-3 months. But the authors have said they will define 'short- term follow up' as less than one month. This does not make sense in relation to the primary outcome measure. It would make more sense to me that short term was less than 3 months if you are using glycaemic control as a primary outcome. 	The timing of the outcome measure has been modified to read: short-term (≤ 3 months of the intervention period), medium-term (3 to 6 months of the intervention period, and long-term (≥ 6 months after the intervention)	8, 12
Search strategy for the identification of studies:  o Do you mean "key terms" not key concepts"? o Is there any reason you have excluded Google Scholar and Web of Science from the list of databases? The authors state that a limitation of previous reviews is the limited databases searched so it is important that authors ensure that have overcome this by including all relevant databases. Considering the types of journals that mHealth studies are typically published in, many of which are not indexed in places like Scopus, google scholar may be an important source of studies for this review. 	"Key concepts" is changed to "key words" More databases including Google Scholar and Web of Science will be searched.	7, 8
Table 2:  o Intervention: What about mHealth + usual care? o Comparison: see previous comment above regarding whether you will include control apps 	The statement is modified to read: studies on self-management of T2DM that utilised mHealth apps alone, mHealth apps along with usual care or along with a range of other technologies such as a wearable device (e.g. pedometer) or mHealth apps in conjunction with other mHealth solutions such as texting/messaging.	6, 8
Quantitative synthesis: How do the authors plan to deal with the 2 different measurement of the primary outcome? HbA1c and FBG? How will you compare studies that use the different outcome measures?	All FBG measurements will be converted to an estimated HbA1c value.	7

1. Cui M, Wu X, Mao J, Wang X, Nie M. T2DM Self-Management via Smartphone Applications : A Systematic Review and Meta- Analysis. PLoS One [Internet]. 2016;11(11):1–15. Available from: <https://www.ncbi.nlm.nih.gov/pmc/articles/PMC5115794/>

2. Liang X, Wang Q, Yang X, Cao J, Chen J, Mo X, et al. Effect of mobile phone intervention for diabetes on glycaemic control: a meta-analysis. Diabet Med [Internet]. 2011;28(4):455–63. Available from: <http://www.ncbi.nlm.nih.gov/pubmed/21392066>

3. Frazetta D, Willet K, Fairchild R. A systematic review of smartphone application use for type 2 diabetic patients. *Online J Nurs Informatics* [Internet]. 2012;(2008):1–9. Available from: <http://ojni.org/issues/?p=2041>
4. Khorsan R, Crawford C. How to assess the external validity and model validity of therapeutic trials: a conceptual approach to systematic review methodology. *Evid Based Complement Alternat Med* [Internet]. 2014 [cited 2018 Jul 24];2014:11. Available from: <https://www.ncbi.nlm.nih.gov/pmc/articles/PMC3963220/>
5. Higgins JP., Green S, editors. *Cochrane Handbook for Systematic Reviews of Interventions* Version 5.1.0 [updated March 2011]. The Cochrane Collaboration [Internet]. John Wiley & Sons, Ltd.; 2011. Available from: www.cochrane-handbook.org
6. Moher D, Schulz K., Altman D. The CONSORT statement: revised recommendations for improving the quality of reports of parallel-group randomised trials. *Lancet* [Internet]. 2001 [cited 2016 Oct 27];357:1191–4. Available from: <http://www.thelancet.com/pdfs/journals/lancet/PIIS0140673600043373.pdf>
7. Pal K, Eastwood S V, Michie S, Farmer A, Barnard ML, Peacock R, et al. Computer-based interventions to improve self-management in adults with type 2 diabetes: a systematic review and meta-analysis. *Diabetes Care* [Internet]. 2014 Jun 1 [cited 2018 Jul 25];37(6):1759–66. Available from: <http://www.ncbi.nlm.nih.gov/pubmed/24855158>

VERSION 2 – REVIEW

REVIEWER	Sheyu Li West China Hospital, Sichuan University, China
REVIEW RETURNED	19-Jan-2019

GENERAL COMMENTS	Thanks very much for the careful response and revision by the authors. However, I still have concerns with the response.  1. It is arbitrary to state this systematic review to be the first systematic review (may be the first published protocol) evaluating the interacting components of self-management of type 2 diabetes. Hou C et al. (Diabetes Care. 2016 Nov;39(11):2089-2095) tested the age and duration of diabetes using subgroup analyses. Wu Y et al. (JMIR Mhealth Uhealth. 2017 Mar 14;5(3):e35) and Bonoto BC et al. (JMIR Mhealth Uhealth. 2017 Mar 1;5(3):e4) explored the design and functions of the apps. 2. As responded by the authors, the subgroup analysis may be the most important advantage of the systematic review. However, there are eight predefined subgroup analyses, some of which may introduce more than one degree of freedom. According to the credibility checklist of the subgroup analyses (BMJ. 2012 Mar 15;344:e1553; BMJ. 2011 Mar 28;342:d1569), the number of subgroup analyses could be too many to be credible. 3. It is also not appropriate to blame a systematic review for the absence of the assessment using the CONSORT checklist, which is not necessary in most cases.
---

VERSION 2 – AUTHOR RESPONSE

Reviewer's Comment	Author Response	Page No
It is arbitrary to state this systematic review to be the first systematic review (may be the first published protocol) evaluating the interacting components of self-management of type 2 diabetes. Hou C et al. (Diabetes Care. 2016 Nov;39(11):2089-2095) tested the age and duration of diabetes using subgroup analyses. Wu Y et al. (JMIR Mhealth Uhealth. 2017 Mar 14;5(3):e35) and Bonoto BC et al. (JMIR Mhealth Uhealth. 2017 Mar 1;5(3):e4) explored the design and functions of the apps.	Thank you for the comment. The statement about being the first systematic review evaluating the interacting components of self-management of type 2 diabetes has been modified to read: “To our knowledge, this is the first published protocol that describes how a systematic review will be conducted to evaluate the impact mHealth apps might have on self-management of T2DM”.	14
As responded by the authors, the subgroup analysis may be the most important advantage of the systematic review. However, there are eight predefined subgroup analyses, some of which may introduce more than one degree of freedom. According to the credibility checklist of the subgroup analyses (BMJ. 2012 Mar 15;344:e1553; BMJ. 2011 Mar 28;342:d1569), the number of subgroup analyses could be too many to be credible.	Thank you for your comment on subgroup analysis. We have reduced the number of planned subgroup analyses from eight to three (ethnicity, comorbidities, and behaviour change models used). We also acknowledge that there are scepticisms about the credibility of subgroup effects (1–3). Therefore, we will ensure that subgroup analyses are conducted majorly if the primary outcome of any included trial shows statistically significant differences between intervention groups. Where a trial reports differences in treatment outcome between intervention groups but fails to demonstrate any statistical significance, subgroup analyses will only be carried out to generate hypotheses (3).	12, 13 – 14
It is also not appropriate to blame a systematic review for the absence of the assessment using the CONSORT checklist, which is not necessary in most cases.	Thank you for your comment. We intend to use the CONSORT checklist to judge the reliability or relevance of RCTs included in our review, not in previous systematic reviews (4). Hence, the statement has been modified to read: “The Consolidated Standards of Reporting Trials (CONSORT) checklist will be used to judge the reliability or relevance	6

	of the findings of RCTs that will be included in this review".	
--	--	--

References

1. Assmann SF, Pocock SJ, Enos LE, Kasten LE. Subgroup analysis and other (mis)uses of baseline data in clinical trials. *Lancet* [Internet]. 2000 Mar 25 [cited 2019 Mar 22];355(9209):1064–9. Available from: <http://www.ncbi.nlm.nih.gov/pubmed/10744093>
2. Sun X, Briel M, Walter SD, Guyatt GH. Is a subgroup effect believable? Updating criteria to evaluate the credibility of subgroup analyses. *BMJ* [Internet]. 2010 Mar 30 [cited 2019 Mar 22];340:c117. Available from: <http://www.ncbi.nlm.nih.gov/pubmed/20354011>
3. Sun X, Briel M, Busse JW, You JJ, Akl EA, Mejza F, et al. Credibility of claims of subgroup effects in randomised controlled trials: systematic review. *BMJ* [Internet]. 2012 Mar 15 [cited 2019 Mar 20];344:e1553. Available from: <https://www.bmj.com/content/344/bmj.e1553.long>
4. Moher D, Schulz K., Altman D. The CONSORT statement: revised recommendations for improving the quality of reports of parallel-group randomised trials. *Lancet* [Internet]. 2001 [cited 2016 Oct 27];357:1191–4. Available from: <http://www.thelancet.com/pdfs/journals/lancet/PIIS0140673600043373.pdf>